

# Nickel stress-tolerance in plant-bacterial associations

Veronika Pishchik[1,2], Galina Mirskaya[2], Elena Chizhevskaya[1], Vladimir Chebotar[1] and Debasis Chakrabarty[3]

[1] All-Russia Research Institute for Agricultural Microbiology, Saint-Petersburg, Pushkin, Russian Federation
[2] Agrophysical Scientific Research Institute, Saint-Petersburg, Russian Federation
[3] CSIR-National Botanical Research Institute, Lucknow, India

## ABSTRACT

Nickel (Ni) is an essential element for plant growth and is a constituent of several metalloenzymes, such as urease, Ni-Fe hydrogenase, Ni-superoxide dismutase. However, in high concentrations, Ni is toxic and hazardous to plants, humans and animals. High levels of Ni inhibit plant germination, reduce chlorophyll content, and cause osmotic imbalance and oxidative stress. Sustainable plant-bacterial native associations are formed under Ni-stress, such as Ni hyperaccumulator plants and rhizobacteria showed tolerance to high levels of Ni. Both partners (plants and bacteria) are capable to reduce the Ni toxicity and developed different mechanisms and strategies which they manifest in plant-bacterial associations. In addition to physical barriers, such as plants cell walls, thick cuticles and trichomes, which reduce the elevated levels of Ni entrance, plants are mitigating the Ni toxicity using their own antioxidant defense mechanisms including enzymes and other antioxidants. Bacteria in its turn effectively protect plants from Ni stress and can be used in phytoremediation. PGPR (plant growth promotion rhizobacteria) possess various mechanisms of biological protection of plants at both whole population and single cell levels. In this review, we highlighted the current understanding of the bacterial induced protective mechanisms in plant-bacterial associations under Ni stress.

## INTRODUCTION

Heavy metals (HM) contaminant of agricultural land and water causes major environmental and human health problems (*Roy & McDonald, 2015*; *Ihedioha, Ukoha & Ekere, 2017*). Nickel (Ni) has been indicated as one of the most dangerous HM for the environment, and Ni affected plants undergo a severe stress condition (*Hussain et al., 2013*; *Pietrini et al., 2015*).

Ni enters the soil through a variety of sources such as metal smelters, industrial effluents, Ni- oxide nanoparticles during the manufacture of electronic devices and catalysts, wastewater, including uses of fertilizers and pesticides (*Song et al., 2008*; *Cabanillas et al., 2012*). Among heavy metals, Ni is characterized by barrier-free penetration into the aboveground organs of plants (*Fabiano et al., 2015*; *Deng et al., 2018*). Ni induces cytotoxic and genotoxic effects on plants (*Magaye & Zhao, 2012*; *Manna & Bandyopadhyay, 2017*).

Corresponding author
Veronika Pishchik,
veronika-bio@rambler.ru

Excessive accumulation of Ni in plants leads to oxidative stress, accompanied by an increase in the accumulation of ROS (reactive oxygen species) (*Sharma & Dietz, 2009*), an inhibitor of growth, mineral nutrition (*Parida, Chhibba & Nayyar, 2003*), photosynthesis (*Prasad, Dwivedi & Zeeshan, 2005*), membrane functions, carbohydrate metabolism (*Seregin & Kozhevnikova, 2006*) and water regime of plants (*Llamas & Sanz, 2008*).

Plants have a number of potential mechanisms to protect against high concentrations of heavy metals, with which they manage to survive under metal stress (*Schützendübel & Polle, 2002*). Resistance to heavy metal toxicity depends on reduced absorption, increased in vacuolar sequestration and enhanced expression of defense proteins. There are some recent reviews about strategies adapted by plants to neutralize and overcome the Ni stress (*Fabiano et al., 2015*; *Sachan & Lal, 2017*; *Shahzad et al., 2018b*; *Deng et al., 2018*) that indicate a great interest among scientific community.

PGPR (Plant Growth Promotion Rhizobacteria), inhabiting plant rhizosphere and rhizoplane and interacting with root exudates and soil microbial communities, form strong associations with plants (*Brencic & Winans, 2005*). PGPR in native ecosystems play a key role in protecting plants from various stress factors, including high concentrations of HM. Under HM stress sustainable plant-microbial associations are formed for the joint survival of both partners (*Jing, He & Yang, 2007*). Such plant-bacterial associations were described for plants: Ni-hyperaccumulators *Alyssum bertolonii* (*Mengoni et al., 2001*) and *Tlaspi goesingense* (*Idris et al., 2004*) with their Ni-resistant dominant bacteria from genera *Pseudomonas*, *Methylobacterium*, *Rhodococcus* and *Okibacterium*. Also, such associations are artificially formed when PGPR are used for phytoremediation, the success of which depends on the correctly used association. Endophytic bacteria penetrate into the root cortex, live inside the plant roots in the root cells, improved nutrient uptake and plant growth. Endophytic bacteria are also successfully used for phytoremediation (*Jan et al., 2019*; *Naveed et al., 2020*). Phytoremediation is a green strategy that uses hyper-accumulator plants and their rhizospheric microorganisms to stabilize, transfer or degrade pollutants in soil, water and environment. This technology is considered as well-efficient, cheap and adaptable with the environment (*Nedjimi, 2021*).

This review focuses on plant defense mechanisms in plant-bacterial associations under Ni stress. Previous reviews touched this topic a little (*Pishchik et al., 2016*; *Egamberdieva, Abd-Allah & Silva, 2016*), focusing mainly on other heavy metals, specifically on Pb, Cd, and Zn.

## SURVEY METHODOLOGY

Initially we analyzed the closed-relation reviews to our topic, to choose the general direction and novelties of our review. The key words for each section of review were selected and used in search of Web of Science and Google Scholar databases. To collect all relevant information we used the following keyword combinations: nickel (Ni) contamination and soil; Ni stress and plants; Ni and physiological destructions in plants; Ni stress and plant antioxidant enzymes; Ni and phytochelatins, Ni and metallothioneins; Ni and proline; Ni and salicylic acid; Ni and phytohormones; Ni- stress and plant genes; Ni-stress and

bacterial genes; Ni- stress and microbial community; Ni- stress and PGPR; Ni-stress and endophytic bacteria; Ni stress and plant—bacterial association; Ni and bioremediation. The abstracts of selected papers were initially screened according to plan of our review, which were focused on topics of survival of plant- bacterial systems under Ni stress. We have mostly focused on the papers of 2000–2020, since the main research in this area were done at this time.

## Ni in soil, in plants and in bacterial cells

The contamination of soil with Ni and other HM resulting due to wastes from heavy industry and nonferrous metallurgy is a major environmental concern. However, Ni can be introduced into the environment with mineral fertilizers, waste water, sludge, oil spills and household rubbish (*Ghosh & Singh, 2005*; *Kozlov, 2005*; *Vodyanitskii et al., 2011*). The worldwide average concentration of Ni in natural soils is 22 mg kg$^{-1}$ (*Kabata-Pendias, 2000*). The range of nickel (Ni) concentrations may reach 200–26,000 mg kg$^{-1}$ in polluted soils, as compared with natural soils (10–1,000 mg kg$^{-1}$) (*Sreekanth et al., 2013*; *Yusuf et al., 2011*). The maximum recorded nickel concentration in contaminated soils was observed in Canada and reached 26,000 mg kg$^{-1}$ (*Kabata-Pendias, Mukherjee & Arun, 2007*). The content of Ni in polluted soils of the city exceeds Tentative Permissible Concentrations level by 10–86 folds (*Vodyanitskii et al., 2011*) and 75 times higher with severe pollution compared to background (*Evdokimova, Kalabin & Mozgova, 2011*).

Ni is the main pollutant of farmlands in south and central China (*Rizwan et al., 2017*), the Ni concentrations in China soils may increase in 6.5 times compared to background (*Ding et al., 2008*). Soils of agricultural land near industrial areas in India contain 47 to 178 mg kg$^{-1}$ Ni (*Rajindiran et al., 2015*). In this regard, the problem of the accumulation of excess Ni in agricultural products arises.

Ni is an essential trace element (in low concentrations 0.01–5 µg/g dry weight) for plants. Ni (II) is a functional component in urease (*Gerendas, Zhu & Sattelmacher, 1998*), glyoxalases, peptide deformylases, methyl Co-M reductases, hydrogenases and superoxide dismutases (*Aziz, Gad & Badran, 2007*). However the high level of Ni concentrations (more than $10^{-4}$ M/l) can lead to toxicity symptoms and growth inhibition in most plants (*Hall, 2002*). The average Ni content in wheat leaves (plant-excluder, accumulates metals in roots) is 0.34 mg kg$^{-1}$ (*Kabata-Pendias, Mukherjee & Arun, 2007*). The family *Brassicaceae* has the highest number of hyper accumulator plants (*Reeves & Baker, 2000*). Such representatives of the family as *Alyssum caricum* (Dudley) and *Thlaspi oxyceras* (Boiss.) can accumulate significant amounts of Ni (up to 12,273 and 13,778 mg/kg, respectively) in the leaves (*Shahzad et al., 2018b*).

Ni is an essential component of bacterial enzymes which catalyze metabolic reactions with molecular hydrogen, nitrogen, carbon monoxide and carbon dioxide and is involved in pathogenesis and detoxifications processes (*Mulrooney & Hausinger, 2003*). However, influence of high concentrations of Ni in bacterial cells leads to oxidative stress resulting in damage of nucleic acids, lipid peroxidations, and enzymes inactivation (*Eitinger et al., 2005*).

## Physiological and biochemical destructions caused by Ni in plants

Metal toxicity leads to molecular changes in plants such as: (a) formation of reactive oxygen species (ROS) by auto-oxidation and Fenton reaction (*Noctor, Reichhel & Foyer, 2018*); (b) locking of main functional groups in biomolecules; (c) expulsion of main metal ions from biomolecules (*Gajewska & Skłodowska, 2007*). Ni ions in high concentrations have a destructive effect on growth, mineral nutrition, photosynthesis (*Zaid et al., 2019*) carbohydrate transport and water relations (*Seregin & Kozhevnikova, 2006*). The increasing levels of Ni stress enhanced methylglyoxal, electrolyte leakage, hydrogen peroxide, and lipid peroxidation content in plants (*Zaid et al., 2019*). Ni decreases seed germination and seedling growth due to change in the activity of hydrolytic enzymes, followed by a delay in the transportation of mobilized reserves from endosperm to the embryonic axis (*Ashraf et al., 2011*). Ni may disrupt the membrane stability (*Shahzad et al., 2018b*) by reducing the uptake of Ca and Zn (*Taiz & Zeiger, 2006*). High concentration of Ni in plants leads to mitotic abnormalities, chromosomal aberrations and decrease in the rate of cell stretching (*Sreekanth et al., 2013*; *Manna & Bandyopadhyay, 2017*).

It has been reported that Ni stress can reduce cytosine methylation levels in clover and hemp and the decrease in methylation depends upon the dosage of the heavy-metal stress. Methylation-sensitive amplification polymorphism (MSAP) data shows that the methylation patterns of different plants within the CCGG sites are similar before and after HM stress, suggesting that the stress-induced changes in methylation are not distributed randomly (*Aina et al., 2004*).

The differential regulation of chloroplastic heat shock protein (Cp-sHSPs or HSP26.13p) in *Chenopodium album* protects the plant both from heat and Ni as well as other (Cu and Cd) HM stresses (*Haq et al., 2013*). It was revealed in proteome analysis of different plant species that ubiquitin activity can be reduced significantly by Ni and other HM (Cd, Pb, Co, Cu, Cr, Hg) at 100 $\mu$M concentration, whereas low concentrations can induce 26S proteasome activity. Although these metals induce the accumulation of ubiquitin conjugated proteins, the abundance of 20S core protein in UPS system is not changed (*Aina et al., 2007*; *Pena et al., 2008*).

## Plants defense system

Plants have different levels of protection against elevated levels of HM. The first level is the physical barrier, represented by various morphological structures, such as a thick cuticle, cell walls, trichomes (*Hall, 2002*; *Fourati et al., 2016*). Trichomes can secrete various secondary metabolites to detoxify heavy metals (*Hauser, 2014*). High concentrations of Ni absorbed into the vacuole, which protects the cytoplasm from the toxic effect (*Krämer et al., 2000*). The sequestration of Ni into the leaf vacuole can be connected with a vacuolar metal-ion transporter protein (TgMTP1) (*Persans, Nieman & Salt, 2001*). The second level of biochemical protection is the inclusion of heavy metals into plant tissue. In this case, the plant synthesizes various substances of an enzymatic and non-enzymatic nature (*Manoj et al., 2020*). The response of plants to Ni stress depends on the plant species. At the same time, intraspecific and interspecific hybrid differences in the presence of Ni in high concentrations are noted (*Amjad, 2020*; *Amjad et al., 2020*).

Under stress of high Ni concentrations, plants trigger numerous adaptive mechanisms to neutralize its action, including the induction of many low molecular weight protein chelators, such as phytochelatins and metallothioneins, specific amino acids, such as proline, and activation of antioxidant enzymes (*Dalvi & Bhalerao, 2013*; *Viehweger, 2014*). The equilibrium between the synthesis and detoxification of free radicals in plants is supported by plant enzymes and antioxidants of nonenzymatic nature, such as ascorbate, glutathione, tocopherol, carotenoids and phenols (*Mittler et al., 2004*).

Glutathione (GSH) plays a significant role in cellular redox balance by binding to Ni and other HM. It was found that elevated GSH concentration driven by constitutively elevated SAT (serine acetyltransferase activity) correlated with increased resistance to Ni stress in *Thlaspi goesingense* (*Freeman et al., 2004*). Besides, it was shown that plants resistance to HM is clearly linked to the efficiency of glutathione S-transferases (GST) in the detoxification process (*Wu et al., 2019*). So, for Ni a negative correlation between GST/peroxydase activities and chlorophyll (Chl) content has been indicated (*Helaoui et al., 2020*).

Plants synthesize proline in response to nickel stress. That was shown for various plant species, such as *Triticum aestivum* (*Gajewska & Skłodowska, 2009*), *Brassica oleracea var. capitata* (*Pandey & Sharma, 2002*), *Pisum sativum* (*Gajewska & Skłodowska, 2005*). Proline functioning as osmolyte is also a defense against Ni toxicity (*Seregin & Kozhevnikova, 2006*).

Proline level was higher in Ni-treated rice plants compared to Cd-treated plants. However, Ni cations in a high concentration (1.0 mM) significantly decreased proline synthesis (*Jan et al., 2019*).

Plants can decrease Ni toxicity, chelating Ni cations with various organic acids. Malic acid synthesis is associated with Ni tolerance in plants such as Ni-hiperaccumulator *Stackhousia tryonii* Bailey (*Bhatia, Walsh & Baker, 2005*), ryegrass, and maize (*Yang et al., 1997*).

Ni-tolerance of the eight different species (*Homalium kanaliense* (Vieill.) Briq., *Casearia silvana* Schltr, *Geissoishirsuta* Brongn. & Gris, *Hybanthus austrocaledonicus* Seem, *Pycnandra acuminata* (Pierre ex Baill.) Swenson & Munzinger (syn *Sebertia acuminata* Pierre ex Baill.), *Geissois pruinosa* Brongn & Gris, *Homalium deplanchei* (Viell) Warb. and *Geissois bradfordii* (H.C. Hopkins) was associated with citric acid (*Callahan et al., 2012*).

***Antioxidant enzymes.*** The antioxidant enzymes: superoxide dismutase (SOD), catalase (CAT), glutathione peroxidase (GSH-Px), guaiacol peroxidase(GPX), peroxiredoxins (Prxs) and enzymes of the ascorbate-glutathione (AsAGSH) cycle, such as ascorbate peroxidase (APX), monodehydroascorbate reductase (MDHAR), dehydroascorbate reductase (DHAR), and glutathione reductase (GR) are indicative enzymes for a high level of abiotic stress in plants (*Mittler et al., 2004*).

The effect of Ni on antioxidant enzymes is different for different types of plants. *Pandey & Sharma (2002)* reported that Ni reduced CAT and POD activities in cabbage leaves. However in pigeon pea (*Cajanu scajan L.*) there was no change in CAT activity under Ni stress, while SOD, glutathione reductase (GR) and POD activities were increased (*Rao & Sresty, 2000*). Ni stress significantly decreased activities of CAT and SOD and increased activities of glutathione peroxidase (GSH-Px) in wheat plants (*Gajewska et al., 2006*;

*Gajewska & Skłodowska, 2007*). In shoots of Ni-stressed *Solanum nigrum* L., an enhanced activity of SOD and APX, accompanied by a decline of CAT activity were observed. In roots, increases in SOD and CAT activities were detected in response to Ni, whilst APX was not increased (*Soares et al., 2016*).

Data on plants antioxidant activities are summarized in Table 1. Since various actions of Ni on plant antioxidant enzymes are described, further study of this direction is necessary.

***Phytochelatines* (PCs)** are the most important metal-binding ligands, since it is believed that the synthesis of these compounds is one of the key detoxification mechanisms (*Chen et al., 2008*). PCs are low molecular weight, short-chain thiol repeating proteins that have high affinity for binding to HMs when they are at toxic levels (*Lee et al., 2002*; *Chen et al., 2008*; *Shukla et al., 2013*). PCs are produced in plants from sulfur-rich glutathione (GSH) using phytochelatin synthase (PCS). PCs form high-molecular complexes with toxic metals, including Ni, in the cytosol and subsequently transfer them to plant vacuoles (*Song et al., 2014*). Induction of PCs synthesis occurs within cells as a result of exposure to various levels of Ni in both the roots and above-ground organs. Nickel accumulation resulted in formation of PCs in *Nicotiana tabacum* L and *Thlaspi japonicum* (*Nakazawa et al., 2001*; *Mizuno et al., 2003*).

PCs' synthesis is considered as one of the protective functions of plants against the stress of nickel and other metals (*Talebi et al., 2019*). It has been suggested, that PCs may serve as a biological marker for Ni accumulation in plants (*Ameen et al., 2019*). That suggestion is confirmed in the study of phytochelatins gene expression in response to the action of different concentrations of nickel in alfalfa plants (*Helaoui et al., 2020*).

Though Ni was a relatively effective activator of PC synthase during *in vitro* studies, a functioning more effective and alternative detoxification mechanisms, such as metallothioneins and histidine in plants was proposed (*Cobbett, 2000*).

***Metallothioneins* (MTs)** are low molecular weight cysteine-rich proteins (4 –8 kDa) that make up an extremely heterogeneous family of metal-binding proteins that are ubiquitous in cells (*Peroza & Freisinger, 2007*). In plants, MTs are involved in neutralizing HM toxicity through cell sequestration, homeostasis of intracellular metal ions, and regulation of metal transport (*Guo et al., 2013*). MTs form metal-thiolate complexes; therefore, they can tolerate to elevated concentrations of metals (*Kumar et al., 2012*; *Mirza et al., 2014*).

Ni increases the MTs expressions in *Solanum nigrum* (*Ferraz et al., 2012*) and *Lupinus luteus* (*Jaskulak et al., 2019*), that results prove the involvement of MTs in Ni homeostasis and detoxification.

It is shown that MTs genes can be used to create HM-resistant plant-microbial systems and their subsequent application in phytoremediation or phytostabilization technologies (*Pérez-Palacios et al., 2017*; *Tsyganov et al., 2020*).

***Phytohormones*** are classified into different groups (auxins, cytokinins, gibberellins, brassinosteroids, salicylic acid, abscisic acid, and jasmonates) and plays different roles in plant growth and development.

**Table 1 Plants defense mechanisms under Ni stress.**

| Plant species | Ni conc. | Mechanisms | Reference |
|---|---|---|---|
| *Triticum aestivum* L. | 25–50 µg/L | (+) SOD activity, proline content | *Gajewska et al. (2006)*, *Parlak (2016)* |
| | 200 µM | (+) proline content in shoots; POD, GST activities (-) SOD, CAT activities | |
| *Atropa belladonna* L. | 50–200 µM | (+) proline, spermine, spermidine contents (-) content of putrescine | *Stetsenko, Shevyakova & Kuznetsov (2011)* |
| *Solanum nigrum* L | 100 µM | (+) SOD and CAT activities in roots (+) SOD and APX activities in shoots (-) CAT activity in shoots | *Soares et al. (2016)* |
| *Oryza sativa* L. | 10–50 µM | (-) MDA concentrations | *Rizwan et al. (2017)* |
| | 100–200 µM | (+) proline content; POD and CAT activities in roots and shoots (-) SOD activity in roots and shoots | |
| *Lactuca sativa* L. | 400–600 mg/kg of soil | (+) CAT, POD, SOD activities in shoots (+) MDA and GST levels | *Zhao et al. (2019)* |
| *Alyssum inflatum* Nyár. | 100–400 µM | (+) proline content; (+) SOD, POD, CAT, APX activities | *Najafi, Karimi & Ghasempour (2019)* |
| *Hydrilla verticillata* (Lf) Royle | 5–15 µM | (+) SOD and CAT activities in leaves and stems; POD activity in leaves (-) POD activity in stems | *Song et al. (2018)*, *Zhang et al. (2020)* |
| | 20–40 µM | (-) SOD and CAT activities in leaves and stems; POD activity in leaves (+) POD activity in stems | |
| *Vigna mungo* L. | 10–100 µM | (+) proline content | *Singh et al. (2012)* |
| *V. cylindrical* L., *V. radiate* L. | 50–150 M | (+) SOD, CAT and POD activities in roots | *Mahmood et al. (2016)* |
| *Populus nigra* L. | 200–800 M | (+) CAT and APX activities in leaves | *Kulac et al. (2018)* |
| *Pisum sativum* L. | 100 M | (+) SOD, POD, CAT, APX, GSH-Px, GR activities, proline, glycinebetaine contents | *Balal et al. (2016)* |
| *Landoltia punctate* | 0.01–0.5 mg/L | (+) SOD, POD, CAT activities | *Guo et al. (2017)* |
| | 5–10 mg/L | (-) SOD, POD, CAT activities | |
| *Grewia asiatica* L. | 20 mg/kg of soil | (-) SOD, CAT, POD activities | *Zahra et al. (2018)* |
| | 40–60 mg/kg of soil | (+) POD activity (-) SOD, CAT activities | |
| *Glycine max* L. | 0.05–20 µM | (+) SOD, POD activities | *Reis et al. (2017)* |
| *Catharanthus roseus* L. | 2.5–50 mM | (+) proline content, CAT activity | *Arefifard, Mahdieh & Amirjani (2014)* |
| *Medicago sativa* L. | 50–500 mg/kg | (+) POD activity | *Helaoui et al. (2020)* |
| *Avéna sativa* L., *Panicum miliaceum* L. | 10–40 ppm | (+) proline content, POD and SOD activities in roots and shoots (-) CAT activity in roots and shoots | *Gupta et al. (2017)* |
| *Amaranthus paniculatus* L. | 25–150 µM | (+) GSH-Px, SOD activities in leaves (-) APX, CAT, GSH-Px, SOD activities in roots | *Pietrini et al. (2015)* |
| *Brassica juncea* L. | 100–400 µM | (+) proline content, SOD activity (-) APX, CAT activities | *Thakur & Sharma (2016)* |

**Notes.**
(+), increased; (-), decreased; APX, ascorbate peroxidase; CAT, catalase; MDA, malone dialdehyde; GSH-Px, glutathione peroxidase; GST, glutathione S-transferase activity; GR, glutathione reductase; POD, peroxidase activity; SOD, superoxide dismutase.

A positive non-significant effect of combined application of gibberellins and cytokinins effect on Ni phytoextraction efficiency of *Alyssum corsicum* was demonstrated (*Cabello-Conejo et al., 2013*). Auxins were found as the most effective phytohormones for increasing Ni yield from Ni hyperaccumulating *Alissum* and *Noccaea* species. All the phytohormones increased plants biomass, but not in all cases the increase in biomass was associated with an increase in nickel yield (*Cabello-Conejo, Prieto-Fernández & Kidd, 2014*).

It was found that application of gibberellins, cytokinins and auxins generally led to a reduction in shoot Ni concentration of *Alissum* and *Noccaea* species (*Cabello-Conejo, Prieto-Fernández & Kidd, 2014*).

The application of epibrassinolide (EBL) recovered the growth *Brassica juncea* and reduced Ni uptake in roots and shoots and improved activities of SOD, CAT, APOX and POD (*Kanwar et al., 2013*), as well as photosynthetic pigments, osmolyte accumulation in *Solanum nigrum* (*Soares et al., 2016*). Application of EBL under Ni stress helps to obtain large plant biomass but possible mechanism of epibrassinolide is still poorly understood (*Shahzad et al., 2018a*).

Abscisic acid (ABA) induces ethylene biosynthesis in adult plants and promotes their senescence and abscission (*Liu et al., 2016*). Under stress conditions, ABA signaling interacts with plants gibberellin and auxin signaling pathways and controls lateral root development (*Zhao et al., 2014*). Ni stress in rice increased the ABA level and ABA was increased with increased heavy metal concentration (*Jan et al., 2019*). Opposite, concentration of salicylic acid (SA) decreased significantly under HM stress, which confirmed the antagonistic effect between SA and ABA (*Jan et al., 2019*).

SA is plant phenolic, and is present in plants as a free and conjugated form (*Maruri-López et al., 2019*). SA can alleviate HM toxicity, decrease ROS, protect membrane stability, interact with other plant hormones, up-regulate hemeoxygenase, improve the performance of the photosynthetic machinery (*Sharma et al., 2020*). SA plays a key role in the regulation of plant growth, development, in defense from HM stress and in plant responses (*Freeman et al., 2005*; *Pasternak et al., 2019*). It is known, that GSH- Glutathione mediated Ni tolerance mechanism in *Thlaspi* hyperaccumulators is signaled by the constitutively elevated levels of salicylic acid (SA) (*Freeman et al., 2005*). SA alleviates metal toxicity influencing their uptake and accumulation in plant organs (*Dalvi & Bhalerao, 2013*). Application of SA under Ni-stress reduced ROS, $H_2O_2$ and MDA contents and lipoxygenase activity, thus up-regulating the capacity of antioxidant defense system in chloroplasts of maize (*Wang et al., 2009*) and wheat (*Siddiqui et al., 2013*), accelerated the restoration of growth processes and improves the total alkaloid content in periwinkle (*Catharanthus roseus* L.) (*Idrees et al., 2013*).

The role of phytohormones in HM stress is discussed in scientific literature but the effects of phytohormones on plants differ and depend on the application rates and time, as well as on the environmental factors and plant species.

## Genes involved in plant protection system to Ni stress

Ni uptake by plant roots can be connected with Fe transporter(s). For instance, in the Ni hyperaccumulator *Alyssum inflatum*, Fe accumulation in roots was stimulated by increased

Ni concentrations (*Ghasemi, Ghaderian & Krämer, 2009*) because of the lack of substrate specificity of AtIRT1. Ni cations could be absorbed *via* the ferrous transporter IRT1 in *A. thaliana* (*Nishida et al., 2011*; *Nishida, Aisu & Mizuno, 2012*).

No specific Ni efflux transporter has been identified. When getting into xylem vessels, Ni transport is mainly driven by leaf transpiration (*Centofanti et al., 2012*). Ni absorption by leaf cells may involve transporters from the ZIP family (ZNT1 and ZNT2), as the gene expression of these transporters triggered under Ni stress (*Visioli, Gulli & Marmiroli, 2014*).

Based on recent genetic studies, the following genes were proposed as candidates for Ni-stress in different plant defense system: serine acetyltransferase (SAT), glutathione reductase (GR) in *Thlaspi goesingense* (*Freeman et al., 2004*), glutathione S–transferase in *Betula papyrifera* (*Theriault, Michael & Nkongolo, 2016*), 1-aminocyclopropane-1-carboxylic acid deaminase (ACC) in *Brassica napus* (*Stearns et al., 2005*) and in *Quercusrubra* (*Djeukam & Nkongolo, 2018*), nicotianamine synthase (NAS3) in *Thlaspi caerulescences* (*Mari et al., 2006*) and *Populus tremuloides* (*Czajka, Michael & Nkongolo, 2018*), thioredoxin family protein in *Chlamydomonas reinhardtii* (*Lemaire et al., 2004*) and in *Betula papyrifera* (*Theriault, Michael & Nkongolo, 2016*)

In recent years, the search for genes responsible for plant resistance to nickel stress became one of the important areas. Several candidate genes that are involved in plant protection against Ni stress have been identified. However, work in this direction should be continued.

There are also very few works concerning the study of the level of expression of plant genes under nickel stress. The main nickel resistance mechanism in *Betula papyrifera* is a downregulation of genes associated with translation (in ribosome), binding, and transporter activities (*Theriault, Michael & Nkongolo, 2016*). Four nicotianamine synthase genes in *Arabidopsis* were upregulated under Ni stress (*Kim et al., 2005*). GS and GOGAT activities were inhibited and the expression levels of their associated genes (*OsGS2, OsFd-GOGAT and OsNADH-GOGAT*) were downregulated in response to Ni stress.

It is known, that microRNAs in plants involved in the post-transcriptional regulation of genes expression and are critical regulators of HM stress (*Dubey et al., 2018*). So miR838 was found as the most responsive to the Ni- stress in *Ricinus communis* L (*Celik & Akdaş, 2019*). However, the role of microRNAs in Ni stress is poorly understood and new information to explore their role is necessary.

## Bacterial genes involved in Ni stress

Ni uptake by microorganisms is regulated by secondary transporters and by ATP-binding cassette (ABC) systems (*Eitinger & Mandrand-Berthelot, 2000*; *Mulrooney & Hausinger, 2003*; *Maitra, 2016*). The secondary systems - nickel/cobalt transporters (NiCoTs; TC 2.A.52.) are widely distributed in bacteria as well as in some archaea and fungi (*Eitinger et al., 2005*). The best investigated ABC-type Ni permease is NikABCDE system of *E. coli*, composed of a periplasmic binding protein (NikA), two integral membrane proteins (NikBC) and two ABC proteins (NikDE). In *E. coli*, Ni overstress is avoided *via* the repressor NikR, which binds to the promoter region of the *nik*ABCDE operon when Ni is present (*De Pina et al., 1999*; *Chivers & Sauer, 2000*). NikR has both strong (in the pM range) and

weak (nM) Ni-binding sites, allowing to detect Ni at concentrations corresponding to the range from 1 to 100 molecules per cell (*Bloom & Zamble, 2004*; *Maitra, 2016*).

HupE/UreJ and UreH are two other families of suspected secondary metal carriers that are distantly related to NiCoTs (*Eitinger et al., 2005*). HupE/UreJ proteins are common among bacteria and encoded within certain hydrogenase (NiFe) or urease gene clusters (*McMillan, Mau & Walker, 1998*; *Baginsky et al., 2004*). Gene *ureH* was found in the urease operon in thermophilic bacteria (*Maeda et al., 1994*). These genes have similar sites to NiCoTs and presumably participate in Ni transport.

Often bacterial nickel resistance is plasmid mediated. For example, in resistant to heavy metals bacteria *Cupriavidus metallidurans* CH34 harbors plasmid pMOL28 which is responsible for Ni, Hg and Cr resistance (*Nies et al., 1989*; *Mergeay, 1985*). Ni efflux driven by a RND transporter is the basis of resistance in this strain. Two operon systems have been studied, a nickel-cobalt resistance Cnr (*cnr*CBA are structural resistance genes with *cnr*YXH regulatory genes) (*Liesegang et al., 1993*) and a Ni-Co-Cd resistance, Ncc (*nccCBA operon*) (*Schmidt & Schlegel, 1989*). The *atm*A gene (encodes ABC-transporter) was also found in the genome, which increases Ni and Co resistance in both *C. metallidurans* and *E. coli* and probably works together with other resistance operons (*Mikolay & Nies, 2009*).

Two distinct Ni resistance loci (*ncc* and *nre*) were found on plasmid pTOM9 from *Achromobacter xylosoxidans* 31A. Expression of the *nre*B gene was specifically induced by Ni and conferred Ni resistance on both *A. xylosoxidans* 31A and *E. coli* (*Grass et al., 2001*). Other resistant gene in *E.coli* is the *rcn*A (*yoh*M) gene responsible for Ni and Co efflux (*Rodrigue, Effantin & Berthelot, 2005*). In the unicellular cyanobacterium *Synechocystis* sp. PCC 6803 and *Helicobacter pylori*, a Ni resistance operon *nrs* and *czn* operon (Cd, Zn and Ni resistance) had been described respectively (*García-Domínguez et al., 2000*; *Stahler et al., 2006*). NrsB and NrsA proteins are homologues to CzcB and CzcA and they probably form a membrane-bound protein complex catalyzing Ni efflux by a proton/cation antiport.

Although bacterial genes involved in the transfer and accumulation of nickel have been studied, some questions remain unclear. For example, there is very little information on the genetic regulation in plant-bacterial associations. Plant-associated bacteria probably have a different genes enabling adaptation to the plant environment. The research in this direction is just emerging.

## Plant-bacterial associations

The rich diversity of root exudates and plant rhizodeposits attract diverse and unique microbial communities (*Brencic & Winans, 2005*; *Chaparro et al., 2012*). In plant-microbial associations, the host plant and associated microorganisms form a multicomponent integral system with new properties determined by the interaction of partners. Rhizobacteria can modulate their metabolism depending on the composition of root exudates towards optimizing nutrient acquisition (*Hardoim, Van Overbeek & Van Elsas, 2008*; *Liu et al., 2019*).

Root exudates and signal compounds that regulate the structure and diversity of the rhizosphere and rhizoplane microbial communities, and indirectly regulate the fluxes of biologically active substances synthesized by microorganisms (*Bais et al., 2006*; *Smith,*
*Gravel & Yergeau, 2017*). Therefore plants can modulate its microflora by dynamically adapting it to the environment (*Vandenkoornhuyse et al., 2015*; *Liu et al., 2019*).

In its turn, rhizobacteria can modulate their metabolism depending on the composition of root exudates towards optimizing nutrient acquisition (*Hardoim, Van Overbeek & Van Elsas, 2008*). PGPR also can absorb ACC excreted from the plants and hydrolyzed by the ACC deaminase decreasing the content of ACC from the environment and consequently reduce stress ethylene level (*Glick, 2005*).

It was found, that *Pseudomonas putida, Pseudomonas fluorescens* can inhabit not only in soil, but on plant leaves and roots and form biofilms (*Ude et al., 2006*). The formation of the biofilm is influenced by the quorum-sensing (QS) process (*Fuqua & Greenberg, 2002*). The mechanisms of quorum formation are described in the review (*Danhorn & Fuqua, 2007*) AHLs (N-acyl-L-homoserine lactones) are the key components of QS signaling system (*Danhorn & Fuqua, 2007*; *Ortíz-Castro et al., 2009*). Plants identify AHLs and trigger changes in gene expression, defense responses of plants (*Ma et al., 2016a*).

Plants also can form in their roots specific symbiotic associations with microorganisms living in the spaces between cells of the root cortex and providing plants with nitrogen (such as plant-rhizobia and arbuscular mycorrhiza). The nitrogen-fixing rhizobia associated with legumes (*Gray & Smith, 2005*; *Djordjevic, Mond-Radzman & Imin, 2015*) as well as mycorrhizal fungi formed a symbiosis with the roots of most vascular plants are well understood (*Upadhyaya et al., 2010*; *Emamverdian et al., 2015*). The value of nitrogen fixation is very high for plants, and it is concluded that nitrogen fixation is of great ecological importance as a way to replenish the nitrogen available to plants in most natural ecosystems (*Djordjevic, Mond-Radzman & Imin, 2015*).

Bacteria, inhabiting in rhizosphere were classified according to their functional activities (*Ahemed, 2019*). The following groups PGPR were allocated: rhizomediators (solubilizing the HM and regulating HM availability), phytostimulators (stimulating plant growth because of phytohormone production), biofertilizers increasing soil nutrient availability, biopesticides (controlling plant pathogens and diseases) (*Ahemed, 2019*). All these properties are important in plant-microbial interactions under HM stress, including biocontrol function, which ensures the systemic resistance of plants (*Ahemed, 2019*; *Manoj et al., 2020*).

Despite a fairly long study of the microbial community of plants, our knowledge about it is quite limited (*Quiza, St-Arnaud & Yergeau, 2015*). Plant-associated bacteria probably have a different genes enabling adaptation to the plant environment. The studying in this direction is just beginning. Recently two sets of plant-associated bacteria genes (involved in plant colonization, and microbial competition between plant-associated bacteria) have been revealed in sequencing 484 bacterial genomes of bacterial isolates from roots of *Brassicaceae*, poplar, and maize (*Levy et al., 2018*). In addition to that 115 genes, which consist of 2% of all genes of *Pseudomonas simiae* (with colonization functions of the root system of *Arabidopsis thaliana*) were identified (*Cole et al., 2017*). A little earlier it has been shown, that wild accessions of *Arabidopsis thaliana* differ in their ability to form associations with *Pseudomonas fluorescens*, which effects on host health (*Haney et al., 2015*).

The new data, concerning plant-bacterial communication in the associations, such as plant-bacterial signaling in bacterial colonization of plant, quorum-sensing and biofilm formation, both in natural conditions and under Ni stress, increase our knowledge of plant-bacterial associations.

## Bacterial defense systems and PGPR mediated plant defense strategies

PGPR have developed some strategies to eliminate the inhibitory effects of HM toxicity (*Qian et al., 2012*; *Babu, Kim & Oh, 2013*; *Ma et al., 2016a*; *Tiwari S. Lata, 2018*). These strategies are speculated may be summarized schematically in (1) HM biosorption/precipitation by cell surface; (2) HM efflux pumping out of the cell by the transporter system; (3) HM binding in cell vacuole and other intracellular compartments; (4) exclusion of HM chelates into the extracellular space; and (5) enzymatic redox reaction *via* conversion of HM cations into a less toxic state. However, detoxification mechanisms are highly affected by the bacterial species and strains (*Aktan, Tan & Icgen, 2013*). Herewith one strain can simultaneously possess multiple defense mechanisms (*Choudhury & Srivastava, 2001*).

*Glutathione,* intracellular polyphosphate granules, low and high molecular weight proteins and polyoxybutyric acid are involved in the defense system of bacteria when HM is absorbed by bacteria. However, the main defense mechanisms are realized outside bacterial cells, due to a change in the pH and redox potential of the medium, the mobilization of phosphates or the production of polysaccharides, siderophores and various antioxidant enzymes (*Pishchik et al., 2016*). The activity of glutathione-reductase was significantly increased in pea plants, (growing under Ni and Zn stresses) after the inoculation with *Rhizobium* sp. RP5 (*Wani, Khan & Zaidi, 2008*).

*Bacterial extracellular polysaccharides* (EPS) can bind HM (*Ahemad & Kibret, 2013*), these substances can form complex with HM or by forming an effective barrier surrounding the cell (*Rajkumar et al., 2010*). Bacterial biofilms also may take part in sequestration or accumulation of Ni, and other HM, such as Al, Cd, Cu, Cr, Mn, Pb, Se, Zn (*Khan et al., 2016*). Endophytic bacterium *Caulobacter* sp. MN13 (alone and in combination with zeolite) reduced Ni uptake by sesame plants due to bacterial EPS and improved biochemical and agronomic parameters of plants (*Naveed et al., 2020*).

PGPR also produce a specific mixture of *VOC*s (volatile organic compounds) that modulates plant growth hormones and plays important roles in their interactions with plants (*Raza & Shen, 2020*). It was shown that the rice inoculation with *Klebsiella variicola* F2, *Pseudomonas fluorescens* YX2 and *Raoultella planticola* YL2 lead to accumulation of GB (N,N,N-trimethyl glycine) and its precursor choline and improved water content in leaves (*Gou et al., 2015*). GB *in vivo* is both an effective osmoprotectant and a compatible solute (*Felitsky et al., 2004*). It was found also, that GB increased under Ni-stress (*Sirhindi et al., 2016*).

PGPR can secrete *low molecular weight organic acids* which increase Ni and other HM bioavailability for plant uptake (*Becerra-Castro et al., 2011*; *Almaroai et al., 2012*). A number of organic acids such as, citric, oxalic, malonic lactic etc. have chelating properties

(*Panhwar et al., 2013*). The salts formed from these organic acids with heavy metals enter the plants. So, Ni-gluconate and Ni-citrate complexes were found to be present in the cocoa (*Peeters et al., 2017*). The potential of organic acids producing PGPR was highlighted in review (*Rajkumar et al., 2012*); however there is a controversial study, which did not show significant effect on the mobilization of HM (*Evangelou, Ebel & Schaeffer, 2006*; *Park et al., 2011*). This effect probably was attributed by increasing rhizosphere soil pH, or the presence of base cation saturations, which can decreased HM availability, as it was shown for Ni cations (*Giovannetti et al., 2020*).

*Biosurfactants* are classified based on their biochemical nature or the producing microbial species. These natural compounds are classified into five major groups liposaccharides, lipopeptides, phospholipids, fatty acids (and neutral lipids), glycolipids (*Sarubbo et al., 2015*). PGPR strains from the genera *Acinetobacter*, *Pseudomonas* and *Bacillus* are found produced biosurfactants such as alasan, emulsan, glycolipid biosurfactant and surfactin (*Sarubbo et al., 2015*).

It was suggested, that biosurfactant molecules play a key role towards development and maintaining biofilms due to maintenance of water channels through the biofilm (*Banat, De Rienzo & Quinn, 2014*). Biosurfactants have been successfully employed in the remediation of environments contaminated with heavy metal ions (*Sarubbo et al., 2015*), *i.e.,* the lipopeptide biosurfactants from *Bacillus subtilis* A21 bound significant quantity of HM, including 32% Ni of polluted soil and was proposed for remediation (*Singh & Cameotra, 2013*).

*Siderophores.* Because Fe (II) is highly toxic in its free form due to its participation in the Fenton reaction, and Fe (III) is insoluble in solutions and not bioavailable (*Miethke & Marahiel, 2007*), the microorganisms have developed an iron absorption strategy through siderophores (*Rajkumar et al., 2012*; *Ashraf et al., 2017*; *Deicke et al., 2019*; *Hofmann, Morales & Tischler, 2020*). Siderophore is also reported to suppress the plant pathogens in different plants, such as tomato (*Aznar & Dellagi, 2015*), pepper (*Yu et al., 2011*) and maize (*Pal et al., 2001*) due to its participation in plants induce systemic resistance (ISR) (*Bakker, Pieterse & Van Loon, 2007*; *Ghosh, Bera & Chakrabarty, 2020*). Siderophores are low molecular - weight metabolites (500–1,500 daltons) with high affinities for $Fe^{3+}$ with stability constants (*Andrews, RobinsonA & Rodríguez-Quiñones, 2003*). Depending on the functional group siderophores are generally classified in different groups: hydroxamates, catecholates (including phenolates), carboxylates and mixed type siderophores (*Hofmann, Morales & Tischler, 2020*). However, their structural nature is variable and they bind different metals and even metalloids (*Retamal-Morales et al., 2018*). Such complexes are transported into the periplasm by TonB-dependent transporters (TBDT), and are transported across the plasma membrane by ATP-binding cassette (ABC) transporters in both Gram-negative and Gram-positive bacteria (*Ghssein & Matar, 2018*). Other metallophores are found and described also for their ability to uptake metals other than iron, such as, for example, nickelophore for nickel (*Lebrette et al., 2015*), and zincophore for zinc (*Bobrov et al., 2017*). However, the complexes with Ni are stable, compared to complexes with Fe (*Hofmann, Morales & Tischler, 2020*). Bacteria can produce more than one siderophore, so *Pseudomonas aeruginosa* produces pyoverdine and

pyochelin (*Minandri et al., 2016*). The new metallophor pseudopaline from *Pseudomonas aeruginosa* is known more specific for the chelation of nickel and zinc (*Lhospice et al., 2017*).

In addition to the main function of supplying microorganisms and plants with iron, other functions of siderophores are described in the literature. We will focus on the most interesting ones concerning the bacterial protective properties against HM stress. The signaling function of siderophores is discussed in literature (*Roux, Payne & Gilmore, 2009*; *Johnstone & Nolan, 2015*). It is suggested, that the siderophore itself, or a metal complex thereof, acts directly as a signaling molecule or a mediator of quorum sensing (*Roux, Payne & Gilmore, 2009*; *Dembitsky, Quntar & Srebnik, 2011*).

PGPR, producing siderophores, generally increase HM bioavailability through complexation reactions (*Khan et al., 2017*; *Sujkowska-Rybkowska et al., 2020*). Therefore such PGPR can be used in phytoremediation to improve the phytoextraction of Ni and other HM. It was found that more than 80% of endophytic bacteria increased the production of siderophores in the presence of heavy metals (Ni and Co, Cr, Cu, Zn) and also reduced metal toxicity in their host plant *Alyssum bertolonii* (*Ma et al., 2011a*). Ni-resistant *Pseudomonas sp.* A3R3 increased plants biomass as well as Ni accumulation in *Brassica juncea* and *A. serpyllifolium*, due to ACC-deaminase, siderophores, IAA activities (*Ma et al., 2011b*).

However, there are the opposite evidences too, that bacterial siderophores bound Ni cations (so as Pb and Zn), decreased Ni contents in plants and protecting plants against HM toxicity (*Burd, Dixon & Glick, 1998*; *Dimkpa et al., 2008*; *Dimkpa, Weinand & Asch, 2009*; *Tank & Saraf, 2009*), or did not influence on the HM concentrations in plants (*Kuffner et al., 2010*).

The literature analysis allows us to assume that the mechanisms, determining HM uptake by plants with the participation of bacterial siderophores are still remaining unknown. Moreover, there is a need to isolate and analyze new siderophores from different PGPR for the application of these PGPR both in bioremediation and plant protection against biotic stresses.

***Bacterial phytohormones*** The PGPR regulate the nutritional and hormonal balance in plants and induce plant tolerance to stress (*Spence & Bais, 2015*). The phytohormones synthesis in plant rhizosphere is a mechanism of improvement of plant growth under stress (*Etesami, Alikhani & Hosseini, 2015*).

The participation of microbial auxins in changing of plant root morphology is well studied. Microbial phytohormones affect the metabolism of endogenous growth regulators in plant tissue and change root morphology under heavy metal stress conditions. Auxin-producing PGPR reduced the effect of HM stress on physiological processes in plants (*Pishchik et al., 2009*). Auxin- producing *B. megaterium* MCR-8 increased growth, contents of total phenols, flavonoids, and activities of SOD, CAT, POD, and APX in inoculated *Vinca rosea* plants under Ni stress (*Khan et al., 2017*). Although we found very little literature strongly supporting the involvement of PGPR hormones in plant Ni stress management, we can speculate this topic, based on the non-specifically plant reactions on the abiotic stress. Overcoming some of the adverse effects of Ni-stress can help plants cope with

stress in general. For, example, abscisic acid (ABA) is plant hormone regulated of water misbalance in plants by controlling stomatal closure and stress signal transduction pathways (*Kudoyarova, Kholodova & Veselov, 2013*). Therefore we may suggest that ABA-producing PGPR can help plants overcome water misbalance caused by Ni stress.

Cytokinin-producing bacteria *Bacillus subtilis* IB-22 increased auxin production by wheat roots as well as stimulate root exudation of amino acids. Authors proposed that the ability of rhizobacteria to produce cytokinins and thereby stimulate amino acids rhizodeposition may be important in enhancing rhizobacterial colonization of the rhizoplane (*Kudoyarova et al., 2014*). It was shown that cytokinin-producing *B. subtilis* increased root biomass and cytokinin concentration in leaves of *Platycladus orientalis* by 47.5% under water stress. Cytokinin in plant tissue promoted stomatal opening and mitigated some of the harmful effects of water stress (*Liu et al., 2013*). It also have been reported that cytokinin-producing bacteria from genera *Arthrobacter, Azospirillum, Bacillus* and *Pseudomonas* increased proline content in plant tissue of soybean and shoot and root biomass under salt stress (*Naz, Bano & Ul-Hassan, 2009*). The applying of cytokinin-producing PGPR may be useful in plant protection under Ni-stress. *Bacillus megaterium* LZR216 produced cytokinins, changed root morphogenesis of *Arabidopsis thaliana* and regulated the transcriptional level of cytokinin-responsive environmental sensor AHK3/AHK4 in *Arabidopsis thaliana*. The intact cytokinin-signaling pathway is necessary for PGPR -promoted plant growth and root system architecture alteration (*Jianfeng et al., 2017*).

PGPB effects on plants under Ni stress are summarized in Table 2 and in Fig. 1.

## Bacterial effects on plant stress-responsive genes

Bacteria regulate major metal responsible and transporter genes expression (*Manoj et al., 2020*). It was revealed, that the inoculation with endophytic bacteria *Enterobacter ludwigii* SAK5 and *Exiguobacterium indicum* SA22 led to increasing of Ni content in rice plants, however the expression level of stress-responsive genes, such as *OsGST* (glutathione-s transferase), *OsMTP1* (HM transporting), and *OsPCS1* (phytochelatin synthases) level was lower in treated inoculated plants than in treated non-inoculated plants which indicated a decrease in stress levels when inoculated with bacteria (*Jan et al., 2019*). However, the studies of bacterial effect on gene expression in plants under Ni stress are insufficient and further studies are needed. The missing information on micro RNAs mediated by bacterial inoculation under nickel stress is also necessary.

## Bioremediation

Taking into consideration the differences in responses of plants and microorganisms to HM, bioremediation employs two different approaches: phytoextraction and phytostabilization. Phytoextraction implies the use of HM-accumulating plants that can accumulate metals in aboveground organs, which are then utilized. Phytostabilization (the conversion of chemicals to less mobile and active forms) can employ plants with high resistance to HM, localizing HM mainly in the root system (*Chaney & Mahoney, 2014*). Phytoremediation, an emerging technique makes use of plants and their associated microbes to clean up heavy

**Table 2** Effect of PGPB on plants under Ni stress.

| Plant species | Bacteria | Effect | References |
|---|---|---|---|
| *Brassica campestris* L. | *Kluyvera ascorbata* SUD165 | Bacterium displayed ACC deaminase activity, and produced siderophores, and decreased the level of stress ethylene induced by the Ni | *Burd, Dixon & Glick (1998)* |
| *Brassica juncea* L. | *Bacillus subtilis* SJ-101 | Bacterium produced IAA and stimulated of Ni-phytoextraction | *Zaidi et al. (2006)* |
| *Brassica juncea* L. | *Psychrobacter* sp. SRA1, *Bacillus cereus* SRA10 | Bacteria displayed ACC deaminase activity, produced a siderophores and IAA, increased the accumulation of Ni in the root and shoot tissues | *Ma, Rajkumar & Freitas (2009)* |
| *Brassica juncea* L.; *Brassica oxyrrhina* Coss | *Psychrobacter* sp. SRA2 | Bacterium displayed ACC deaminase activity, produced a siderophores and IAA, increased the fresh and dry biomass of the plants | |
| *Brassica juncea* L.; *Ricinus communis* L. | *Pseudomonas* sp. A3R3, *Psychrobacter* sp. SRS8 | Bacteria improved plant biomass production and decreased heavy metal accumulation | *Ma et al. (2015)* |
| *Brassica juncea* L.; *Luffa cylindrical* L.; *Sorghum halepense* L. | *Bacillus megaterium* SR28C | Bacterium alleviated the detrimental effects of Ni by reducing its uptake and translocation to the plants | *Rajkumar, Ma & Freitas (2013)* |
| *Vigna unguiculata* L. | *Streptomyces acidiscabies* E13 | Bacterium produced hydroxamate siderophores and promoted plant growth under Fe- and Ni-stress | *Dimkpa et al. (2008)* |
| *Vinca rosea* L. | *Bacillus megaterium* MCR-8 | Bacterium alleviated Ni stress, increased the accumulation of total phenols, flavonoids and enzymes SOD, CAT, POD, APX, improved phytoextraction | *Khan et al. (2017)* |
| *Oryza sativa* L. | *Enterobacter ludwigii* SAK5, *Exiguobacterium indicum* SA22 | Bacteria increased plant growth parameters under Cd and Ni stress, also enhance glutathione, proline, and sugar content | *Jan et al. (2019)* |
| *Cicer arietinum* L. | *Pseudomonas aeruginosa* SFP1 | Bacterium declined the level of stress markers (proline and APX, SOD, CAT, and GR), as well as with Cr (VI) and Ni uptake by plants | *Saif & Khan (2018)* |
| *Rafanus sativus* L. | *Bacillus* sp. CIK-516 | Bacterium produced IAA, and displayed ACC deaminase activity, and increased Ni uptake by radish | *Akhtar et al. (2018)* |
| *Triticum aestivum* | *Bacillus subtilis* BM2 | Bacterium displayed ACC deaminase activity, produced IAA, siderophores, ammonia. Bacterium increased plant growth parameters under Ni stress, decreased Ni content in plants and decrease SOD, GR and CAT activity | *Rizvi et al. (2019)* |

**Notes.**

ACC deaminase, 1-aminocyclopropane-1-carboxylic acid deaminase; IAA, indole-3-acetic acid; SOD, superoxide dismutase; CAT, catalase; POD, peroxidase activity; APX, ascorbate peroxidase; GR, glutathione reductase.

metals pollutant from soil (*Kumar & Verma, 2018*). In addition phytoremediation is cost effective and is a more sustainable approach for removal of HM (*Ma et al., 2016b*).

*Streptanthus polygaloides* A. Gray, a Ni- hyperaccumulating plant from *Brassicaceae* family was successfully used in phytoextraction. The shoot Ni concentration of *S. polygaloides* averaged 5,300 mg kg$^{-1}$, whereas Ni concentration in soil was of 3,340 mg kg$^{-1}$ (*Nicks & Chambers, 1995*).

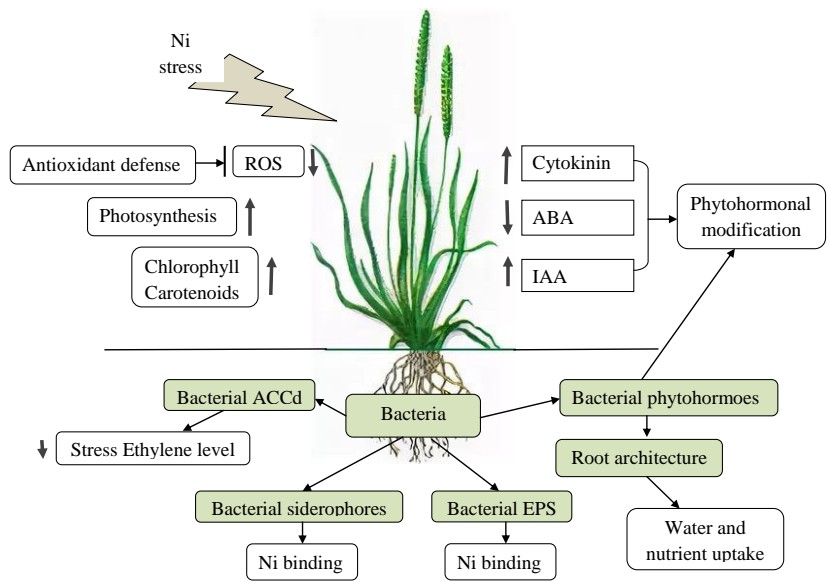

**Figure 1  Effects of PGPB on plants under Ni-stress.**

During development of Ni phytoextraction technology mean Ni concentrations in the shoots *Alissum murale* and *Alissum corsicum* ranged from 4,200 to 20,400 mg kg$^{-1}$. Ni uptake by these Alyssum species was reduced in the field experiments at lower soil pH and increased at higher soil pH, and that was an uninspected result (*Li et al., 2003*). Nowadays eight species of *Alissum* (family *Brassicaceae)* described as Ni-hyperaccumulators (*Pollard, Reeves & Baker, 2014*).

## CONCLUSIONS

The review highlights the importance of bacterial contribution in plant protection under Ni-stress. The understanding of the mechanisms of bacterial plant defense against nickel stress in plant-bacterial associations has been formed in recent decades. Bacteria activated numerous genes in plants in response to Ni- stress; however research in this direction is just emerging. Moreover, intensive study of plant genes involved in protection against nickel stress has also taken place in the last decades, when some plant genes have been identified and proposed as candidates for plant protection against nickel stress. Despite the fact that various mechanisms of bacterial protection have already been described in the literature, some issues remain unexplored. So, more detailed studies of the effect of bacterial phytohormones on plants under Ni stress is required for understanding. More information concerning plant-microbial crosstalk in response to Ni-stress is missing; therefore omics-based technologies, such as transcriptomics, proteomics and metabolomics must be used in future experiments to decipher the mechanisms of bacterial protection of plants. Further study of environmental conditions is also necessary, since the effectiveness of the protective actions of bacteria is also determined by soil conditions and the magnitude of the stress load. The review of the scientific data results on the ability of plant-microbial

associations to regulate the nickel uptake by plants with subsequent utilization of plant biomass will help to develop bioremediation technologies for polluted lands, or to produce eco-friendly agricultural crops on HM contaminated soils (with the level of Ni and other HM contents not exceeding the maximum permissible concentrations).

### Funding

The work of Vladimir Chebotar, Elena Chizhevskaya and Veronika Pishchik was supported by the project "Development of potato breeding and seed production in the Russian Federation" of the Federal scientific and technical program for agricultural development for 2017-2025. The work of Galina Mirskaya was supported by the Ministry of Science and Education of the Russian Federation (Agreement. No. 075-15-2020-805). The funders had no role in study design, data collection and analysis, decision to publish, or preparation of the manuscript.

### Grant Disclosures

The following grant information was disclosed by the authors:
"Development of potato breeding and seed production in the Russian Federation" of the Federal scientific and technical program for agricultural development for 2017-2025.
The Ministry of Science and Education of the Russian Federation: (Agreement No. 075-15-2020-805).

### Competing Interests

The authors declare there are no competing interests.

### Author Contributions

- Veronika Pishchik, Galina Mirskaya, Elena Chizhevskaya, Vladimir Chebotar and Debasis Chakrabarty conceived and designed the experiments, performed the experiments, analyzed the data, prepared figures and/or tables, authored or reviewed drafts of the paper, and approved the final draft.

### Data Availability

There are no raw data or code because this article is a Literature Review.

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
