# Peer review of "Nickel stress-tolerance in plant-bacterial associations"

_PeerJ, doi:10.7717/peerj.12230_

## Round 0.1 · original submission · Major Revisions

Dear Author, this is an interesting piece of work as evaluated by both Reviewers, your manuscript should be however more focused and structured. Please improve your manuscript having in mind the suggestions provided by both Reviewers. Once this is done, you are most welcome to resubmit your manuscript to PeerJ. Good luck with your work!.

Reviewer 1 ·

Basic reporting

This article deals with the Ni stress in plants and plant-microbial interactions under Ni stress. Below are a few points I noted while reading this article:
Literature relating to the Ni stress in plants, and plant-microbial interactions under Ni stress could be expanded
Figures summarizing the establishment of plant-microbial associations under Ni stress could be included to significantly enhance the presentation efficiency

Experimental design

no comment

Validity of the findings

Conclusions could be further revised so as to highlight more details regarding research gaps.

Reviewer 2 ·

Basic reporting

'no comment'

Experimental design

'no comment'

Validity of the findings

'no comment'

Additional comments

The authors of this review made an excellent job of collecting data on plant responses to heavy metal stresses, mainly nickel, especially in the context of plant-microbe interaction. Therefore, this review may be useful for a wide audience including microbiologist, ecologist and plant physiologist at the same time.

Nevertheless, at the current stage this work needs some major changes, it seems to be the preliminary draft for making the review, and not the review itself. Authors have listed numerous information without placing them in the context or drawing some general conclusions. The structure of this work, which consists of many, often very short paragraphs, reflects itself the nature of such a draft. A good review does not just summarize the literature, but discusses it critically, identifies problems, and points out the research gaps and future direction.

The lists of key research questions to be answered should be better highlighted. This review should be more focused on the symbiotic and associative relationships with microorganisms, and interaction with plant growth promoting bacteria under Ni stress, and not the general plant stress responses to Ni.

English language should be improved. It is suggested to use the help of someone with full professional proficiency in English.

---

## Round 0.2 · accepted · Accept

Dear Author, I am happy to inform you that your manuscript is accepted for publication in PeerJ in the current form.

Reviewer 1 ·

Basic reporting

no comment

Experimental design

no comment

Validity of the findings

no comment

Additional comments

The revisions incorporated in the manuscript are commendable, the manuscript may be considered for publication.